# Level-Set Modeling of Grain Growth in 316L Stainless Steel under Different Assumptions Regarding Grain Boundary Properties

**DOI:** 10.3390/ma15072434

**Published:** 2022-03-25

**Authors:** Brayan Murgas, Baptiste Flipon, Nathalie Bozzolo, Marc Bernacki

**Affiliations:** Mines-ParisTech, PSL-Research University, CEMEF—Centre de Mise en Forme des Matériaux, CNRS UMR 7635, CS 10207 Rue Claude Daunesse, CEDEX, 06904 Sophia Antipolis, France; brayan.murgas@mines-paristech.fr (B.M.); baptiste.flipon@mines-paristech.fr (B.F.); nathalie.bozzolo@mines-paristech.fr (N.B.)

**Keywords:** heterogeneous grain growth, anisotropic grain growth, grain boundary energy, grain boundary mobility, finite element method, level-set method, 316L, stainless steel, heterogeneous mobility, anisotropic energy

## Abstract

Two finite element level-set (FE-LS) formulations are compared for the modeling of grain growth of 316L stainless steel in terms of grain size, mean values, and histograms. Two kinds of microstructures are considered: some are generated statistically from EBSD maps, and the others are generated by the immersion of EBSD data in the FE formulation. Grain boundary (GB) mobility is heterogeneously defined as a function of the GB disorientation. On the other hand, GB energy is considered as heterogeneous or anisotropic, which are, respectively, defined as a function of the disorientation and both the GB misorientation and the GB inclination. In terms of mean grain size value and grain size distribution (GSD), both formulations provide similar responses. However, the anisotropic formulation better respects the experimental disorientation distribution function (DDF) and predicts more realistic grain morphologies. It was also found that the heterogeneous GB mobility described with a sigmoidal function only affects the DDF and the morphology of grains. Thus, a slower evolution of twin boundaries (TBs) is perceived.

## 1. Introduction

As most metallic materials exist in the form of polycrystals, determining the kinetics of metallurgical mechanisms such as recovery, grain growth (GG), and recrystallization is crucial, since they determine the final microstructure and properties [1]. Grain boundary (GB) engineering refers to the control of GBs with the aim of obtaining high-performance materials. Thus, numerical models have emerged to help us predict the evolution of microstructures submitted to different thermomechanical loads and the microstructure–property relationship.

The migration of GBs is classically described at the polycrystalline scale by the well-known equation

v=μP, where v is the GB velocity, μ is the GB mobility, and P is the driving pressure. During GG, the evolution of GBs is driven by the reduction of interfacial energy, and the driving pressure is classically defined as a curvature flow driving pressure P=−γκ, where γ is the GB energy and κ is the mean curvature (i.e., the trace of the curvature tensor in 3D). At the polycrystalline scale, this kinematic equation is widely accepted while largely questioned [2,3]. Moreover, a definition of the reduced mobility (μγ) within the misorientation and inclination 5D space is not straightforward [4,5,6,7].

GG has been widely studied at the polycrystalline scale with various numerical approaches such as Phase-Field [8,9,10], Monte Carlo [11,12], Molecular Dynamics [13], Orientated Tessellation Updating Method [14], Vertex [15], Front-Tracking Lagrangian or Eulerian formulations in a Finite Element (FE) context [16,17,18], Level-Set (LS) [19,20,21,22], and Kobayashi–Warren–Carter models [23], to cite some examples. GB energy and mobility have been widely studied since they were reported as being anisotropic by Smith [24] and Kohara [25]. The simplest models use constant values for the GB energy γ and a temperature-dependent mobility, μ(T), which are referred to as isotropic models [8,11,19,26,27]. Heterogeneous models were also proposed, in which each boundary has its own energy and mobility [12,20,21,28,29,30,31,32,33,34,35] trying to reproduce more complex microstructures with local heterogeneity, such as twin boundaries. Each grain has its own crystal orientation, and the GB energy and mobility depend on the disorientation angle between two grains [9,21], but the effect of the misorientation axis and GB inclination is frequently omitted. Thus, general frameworks were proposed, including the GB properties’ dependence on misorientation and inclination [36,37,38], which are categorized as anisotropic models. It must be highlighted that the difference between three-parameter (heterogeneous) and five-parameter (anisotropic) full-field formulations is often unclear in the literature, heterogeneous GB properties being often categorized as anisotropic.

The main reason why most of the studies are carried out using heterogeneous GB properties is the lack of data of GB properties. The early measurements of GB properties (mainly GB reduced mobility) were carried out on bicrystals [39,40,41,42,43,44] leading to the well-known Sigmoidal model [1]. As experimental and computational technologies are improved, new experimental and computational 3D techniques allow studying GG and recrystallization using, for instance, X-ray [45,46,47,48] or molecular dynamics [4,5,49]. Hence, at the mesoscopic scale, few studies have been carried out in 2D using anisotropic GB properties designed by mathematical models [36,37] or by fitting data from molecular dynamics [38]. Nevertheless, these 2D models neglect a part of the 3D space; i.e., the GB inclination is measured in the sample plane, and GB properties are simplified. Finally, regarding the study of GG in 3D, one frequently finds heterogeneous GB properties based on mathematical descriptors of GB properties [33,50,51,52] or based on databases of GB energy values [53,54]. Based on this, two open questions arise: can GB properties be described in 2D using the classical Read–Shockley [55] and Sigmoidal [56] model? Is the effect of anisotropy stronger in 3D? The latter implies carrying out 3D simulations instead of 2D, thus using a better description of GB properties in the 5D GB space.

In a preceding paper [22], four different formulations using an FE-LS approach have been compared. The first is an isotropic formulation used to model different annealing phenomena, such as GG, recrystallization, and GG in the presence of second-phase particles [19,57,58,59,60]. The second is an extension of the isotropic formulation considering heterogeneous values of GB energy and mobility [22]. The third formulation was proposed for triple junctions in [30] and extended to model GG using heterogeneous GB energy in [21] and both heterogeneous GB energy and mobility in [22]. The last one is an anisotropic formulation based on thermodynamics and differential geometry; it was first applied to bicrystals [37] and extended to polycrystals with heterogeneous GB energy and mobility in [22]. In [22], academic cases of triple junctions and polycrystalline microstructures were presented. The main conclusion was that the isotropic formulation can reproduce grain size, mean values, and distributions when the anisotropy level is moderated. However, when the anisotropy level increases, the anisotropic formulation leads to more physical predictions in terms of grain morphology, global surface energy evolution, and multiple junction equilibrium.

The goal of this work is to criticize the capacity of the isotropic and anisotropic formulations to model GG in a real material, which is here a 316L austenitic stainless steel, in terms of mean grain size, grain size distributions, and mean GB properties. We compare the effect of the initial microstructure using statistically representative Laguerre–Voronoï tessellation [61] and digital twin microstructures from EBSD data. The effect of the GB energy definition is illustrated with two different frameworks: a one-parameter, well-known as the Read–Shockley formulation [55]; and a five-parameter one using the the GB5DOF code proposed in [6]. The effect of the GB mobility description using an isotropic and a Sigmoidal model [56] is also discussed. The paper is organized as follows. First, in Section 2, crystallographic definitions, LS treatments, and FE-LS formulations are presented briefly. The methodology to estimate the GB reduced mobility from experimental data is presented in Section 3. Finally, the results using the isotropic and anisotropic formulations are compared using statistically representative Laguerre–Voronoï tessellations (Section 4), immersed microstructures with heterogeneous GB properties (Section 5), and immersed microstructures with anisotropic GB energy using the GB5DOF code [6] (Section 6).

## 2. The Numerical Formulation

The LS method was firstly proposed in [62] to describe curvature flow problems, and it was enhanced later for evolving multiple junctions [63,64] and applied to recrystallization and grain growth in [19,57]. The principle for modeling polycrystals is the following: grains are defined by LS functions ϕ in the space Ω
(1)$$\begin{array}{c} \left\{\begin{array}{c} \phi (X)=\pm d(X,\Gamma ),\;X\in \Omega ,\;\Gamma =\partial G\\ \phi (X\in \Omega )=0\;⇌\;X\in \Gamma , \end{array}\right. \end{array}$$

More precisely, the grain interface Γ is described by the zero-isovalue of the corresponding ϕ function. In Equation (Equation 1), d is the signed Euclidean distance to Γ and ϕ is classically chosen as positive inside the grain and negative outside. The dynamics of the interface is studied by following the evolution of the LS field. When the interface evolution is characterized by a velocity field $\vec{v}$, its movement can be obtained through the resolution of the following transport equation [62]:(2)$$\begin{array}{c} \frac{\partial \phi }{\partial t}+\vec{v}\cdot \vec{\nabla }\phi =0. \end{array}$$

Classically, one LS function is used to describe one grain and Equation (Equation 2) is solved for each grain to describe the grain boundary network evolution. However, when the number of grains, $N_{G}$, increases, one may use a graph coloring/recoloring strategy [59] in order to limit drastically the number of involved LS functions $\Phi =\{\phi_{i},i=1,\cdots ,N\}$ with $N\ll N_{G}$. Two more treatments are necessary. Firstly, the LS functions are generally reinitialized at each time step in order to keep their initial metric property when they are initially built as distance functions to the grain interface, as proposed in Equation (Equation 1):(3)$$\begin{array}{c} ∥\nabla \phi ∥=1. \end{array}$$

In the proposed numerical framework, the algorithm developed in [65] is used. Secondly, the LS evolutions may not preserve the impenetrability/overlapping constraints leading to potential overlaps/voids between grain interfaces at multiple junctions. The solution proposed in [63] and largely used in an LS context [19] is adopted.

The main interest of this global numerical front-capturing framework lies in its ability to define different physical phenomena when they are encapsulated in the velocity field and to deal easily with topological events such as grain disappearance. In the next section, different formulations of the GB velocity and the subsequent FE resolution are presented.

### GB Velocity Formulation

The isotropic formulation uses a homogeneous GB energy and mobility [19]; thus, the velocity field is defined as
(4)$$\begin{array}{c} \vec{v}=-\mu \gamma \kappa \vec{n}, \end{array}$$
where κ is the mean curvature of the boundary in 2D and the trace of the curvature tensor in 3D, and n is the outward unit normal to the boundary. By verifying Equation (Equation 3) and assuming the LS function to be positive inside the corresponding grains and negative outside, the unitary normal and so the mean curvature may be defined as $\vec{n}=-\vec{\nabla }\phi$ and $\kappa =\vec{\nabla }\cdot \vec{n}=-\Delta \phi$, and then, the velocity in Equation (Equation 4) may be written:(5)$$\begin{array}{c} \vec{v}=-\mu \gamma \Delta \phi \vec{\nabla }\phi . \end{array}$$

At the mesoscopic scale, a GB, $B_{ij}$, between grains $G_{i}$ and $G_{j}$, is characterized by its morphology and its crystallographic properties, which may be summarized by a tuple $B_{ij}=\left(M_{ij},n_{ij}\right)$ with two shape parameters describing the interfaces through the unitary-outward normal direction $n_{ij}$ and three crystallographic parameters describing the orientation relationship between the two adjacent grains known as the misorientation tensor $M_{ij}$ (see Figure 1). The misorientation is frequently defined with the axis-angle parameterization, i.e., $M_{ij}\left(a_{i},\theta \right)$, where $a_{i}$ is the misorientation axis and θ is the disorientation [66]. Then, the two quantities of interest, the GB energy γ and GB mobility μ, must be seen as functions from the GB space B to $R^{+}$.

The anisotropic formulation was initially developed using thermodynamics and differential geometry in [37] and was improved in [22] in order to consider heterogeneous GB mobility. Both the GB normal and misorientation are taken into account, and an intrinsic torque term is present:(6)$$\begin{array}{c} v=\mu \left(M\right)\left(P\vec{\nabla }\gamma \left(M,n\right)\cdot \vec{\nabla }\phi -\left({\vec{\nabla }}_{\vec{n}}{\vec{\nabla }}_{\vec{n}}\gamma \left(M,n\right)+\gamma \left(M,n\right)I\right):K\right)\vec{\nabla }\phi \end{array}$$
where I is the unitary matrix, $P=I-\vec{n}⊗\vec{n}$ is the tangential projection tensor, ${\vec{\nabla }}_{\vec{n}}$ is the surface gradient, and $K=\vec{\nabla }\vec{n}=\vec{\nabla }\vec{\nabla }\phi$ is the curvature tensor. The term $\Gamma \left(M,n\right)={\vec{\nabla }}_{\vec{n}}{\vec{\nabla }}_{\vec{n}}\gamma +\gamma I$ is a tensorial diffusion term known as GB stiffness tensor [67,68]. The term $P\vec{\nabla }\gamma \cdot \vec{\nabla }\phi$ in Equation (Equation 6) should be null in the grain interfaces. However, the front-capturing nature of the LS approach, which involves solving Equation (Equation 2) at the GB network and in its vicinity, requires considering this term, which could be non-null around the interfaces. Then, this stabilization term is totally correlated to the front-capturing nature of the LS approach. The term Γ(M,n) is the subject of recent studies for twin boundaries (TBs) Σ3, Σ5, Σ7, Σ9 and Σ11 [67,69]. In the present work, the torque term ${\vec{\nabla }}_{\vec{n}}{\vec{\nabla }}_{\vec{n}}\gamma$ is neglected, but the GB energy still depends on the GB misorientation and inclination, and the kinetic equation could be simplified as:(7)$$\begin{array}{c} v=\mu \left(M\right)\left(P\vec{\nabla }\gamma \left(M,n\right)\cdot \vec{\nabla }\phi -\gamma \left(M,n\right)\Delta \phi \right)\vec{\nabla }\phi . \end{array}$$

Inserting the kinetic Equations (Equation 5) and (Equation 7) into Equation (Equation 2) leads to the weak formulation of the isotropic (Iso) and anisotropic (Aniso) formulation [22]
(8)$$\begin{array}{c} \int_{\Omega }\frac{\partial \phi }{\partial t}\varphi d\Omega +\int_{\Omega }\mu \gamma \vec{\nabla }\varphi \cdot \vec{\nabla }\phi d\Omega -\int_{\partial \Omega }\mu \gamma \varphi \vec{\nabla }\phi \cdot \vec{n}d\left(\partial \Omega \right)=0, \end{array}$$
and
(9)$$\begin{array}{c} \begin{array}{c} \int_{\Omega }\frac{\partial \phi }{\partial t}\varphi d\Omega +\int_{\Omega }\mu \left(M\right)\gamma \left(M\right)\vec{\nabla }\varphi \cdot \vec{\nabla }\phi d\Omega -\int_{\partial \Omega }\mu \left(M\right)\gamma \left(M\right)\varphi \vec{\nabla }\phi \cdot \vec{n}d\left(\partial \Omega \right)+\\ \int_{\Omega }\mu \left(M\right)\left(P\cdot \vec{\nabla }\gamma \left(M\right)+\vec{\nabla }\gamma \left(M\right)\right)\varphi \vec{\nabla }\phi d\Omega +\int_{\Omega }\gamma \left(M\right)\vec{\nabla }\mu \left(M\right)\cdot \vec{\nabla }\phi \varphi d\Omega =0, \end{array} \end{array}$$
respectively.

If the properties are homogeneous, then both formulations are equivalent. The main question is the capability of these two numerical models to reproduce experimental evolution assessed through EBSD data. The next section is dedicated to the parameters identification.

## 3. Parameters Identification

### 3.1. Material Characterisation

The chemical composition of the 316L stainless steel is reported in Table 1. The samples were machined in the form of rectangular parallelepipeds of 8.5 mm × 8.5 mm × 12 mm. Then, the samples were annealed at 1050 °C during 30 min, 1 h, and 2 h. Afterwards, the samples were prepared for EBSD characterization. The preparation consisted of mechanical polishing followed by fine polishing and finally electrolitic polishing; the details of the polishing are listed in Table 2.

Microstructures were analyzed at the center of the sample using a TESCAN FERA 3 Field Emission Gun Scanning Electron Microscope (FEGSEM). It is equipped with several detectors including Symmetry and C-Nano EBSD detectors from the Oxford company. In this work, the Symmetry EBSD detector was used. Post-processing was conducted using the MTEX toolbox in a MATLAB environment [70]. The EBSD map at t = 0 h has a size of 1.138mm×0.856mm and was acquired with a constant step size of 1.5μm. The other three EBSD maps at t=30min,1h, and2h have a size of 1.518mm×1.142mm and were acquired with a constant step size of 2μm. Grain boundaries have a disorientation above 5 degrees (θ> 5°) and Σ3 twin boundaries have a misorientation axis <111>± 5° and θ=60 ± 5°.

The main properties of the initial microstructure are reported in Figure 2. Figure 2b illustrates the grain size and disorientation distribution ignoring Σ3 TBs. The grain size is defined as an equivalent radius, $R=\sqrt{S/\pi }$, where S is the grain area. The microstructure consists of equiaxed grains with an arithmetic mean radius of 15μm, and few bigger grains with a radius around 60μm. Additionally, the microstructure presents a Mackenzie-like DDF typical of random grain orientations [71]. On the other hand, if Σ3 TBs are considered, the DDF presents an additional sharp peak at a disorientation angle θ= 60°, which comes from the TBs and then constitutes a strong source of anisotropy with regard to GB properties (see Figure 2c).

Figure 3, Figure 4 and Figure 5 show the band contrast maps and the grain size distributions at t = 0 s, 30 min, 1 h, and 2 h. Based on Figure 3 and Figure 4, the evolution of the microstructure seems to mostly proceed by normal grain growth (NGG), but the surface grain size distribution shows that the microstrucuture has a bimodal population of grains (see Figure 5). However, some of the grains can reach an equivalent diameter above 0.1 mm, which is much larger than the average grain size.

### 3.2. Estimation of the Average Grain Boundary Mobility Based on the Burke and Turnbull GG Method

In order to compute the average mobility necessary to run full-field simulations, the evolution of the arithmetic mean grain radius ${\bar{R}}_{Nb}$ must be known. Figure 6 shows the evolution of ${\bar{R}}_{Nb}$ as a function of the annealing time. Using the methodology discussed in [72,73], one can obtain an average reduced mobility μγ using the Burke and Turnbull model [74]. This model, where topological and neighboring effects are neglected, is based on five main assumptions: the driving pressure is proportional to the mean curvature, grains are equixaed, the GB mobility and energy are isotropic, the annealing temperature is constant, and no second-phase particles are present in the material. In this context, one can obtain a simplified equation describing the mean radius evolution:(10)$$\begin{array}{c} {\bar{R}}_{Nb}{\left(t\right)}^{2}-{\bar{R}}_{Nb}{\left(t=0\right)}^{2}=\frac{1}{2}\mu \gamma t. \end{array}$$

This methodology has been used in [72,73,75,76,77] assuming general grain boundaries with homogeneous GB energy and mobility. From the evolution of ${\bar{R}}_{Nb}$ in Figure 6 (excluding the Σ3 TBs), one can then obtain a first approximation of the product μγ for the general boundaries at 1050 °C. This approximation will be used for the μγ definition in isotropic simulations. Nevertheless, as illustrated by the second orange curve in Figure 6, when Σ3 TBs are considered in the analysis, grains are of course smaller, but they also grow much slower, with a direct impact on the apparent reduced mobility. This slow evolution can be produced by the strong anisotropy brought by special GBs in the global grain boundary network migration. Different ways to improve the description of the reduced mobility and their impacts in the results are discussed in the following.

In the following sections, *general boundaries* make reference to the case without Σ3 TBs, and the case with Σ3 is referred to as *all boundaries*.

## 4. Statistical Cases

In this section, the 2D GB network is created from the initial experimental grain size distribution shown in Figure 7. The square domain has a length L=2.0 mm, and grains are generated using a Laguerre–Voronoi tessellation [61] based on an optimized sphere packing algorithm [78]. Anisotropic remeshing is used with a refinement close to the interfaces, the mesh size in the tangential direction (and far from the interface) is set to $h_{max}=5\;\mu m$ and in the normal direction $h_{min}=1\;\mu m$, with transition distances set to $\phi_{min}=1.2\;\mu m$ and $\phi_{max}=5\;\mu m$ (see [22,58,79] for more details concerning the remeshing procedure and parameters). The time step is set to Δt=10 s. The orientation field was generated randomly from the grain orientations measured by EBSD in the initial microstructure (Figure 2a). The first part studies grain boundaries without Σ3 TB, (Section 4.1). In the second part, Σ3 TBs are included in the analysis (Section 4.2).

The interfacial energy and average GB properties are computed as
(11)$$E_{\Gamma }=\frac{1}{2}\sum_{i}\sum_{e\in T}l_{e}\left(\phi_{i}\right)\gamma_{e}\;\mathrm{and}\;\bar{x}=\frac{1}{2L_{\Gamma }}\sum_{i}\sum_{e\in T}l_{e}\left(\phi_{i}\right)x_{e},$$
where T is the set of elements in the FE mesh, $l_{e}$ is the length of the LS zero iso-values existing in the element e, i refers to the number of LS functions, $L_{\Gamma }$ is the total length of the GB network Γ, and $x_{e}$ is the GB property of the element e.

### 4.1. Statistical Case with General Boundaries

The first case with general boundaries is composed of $N_{G}=4397$ initial grains. Figure 8a,b show the initial GB disorientation and the initial DDF distribution. Most of the interfaces have a disorientation higher than 15° due to the random generation of orientations that leads to a Mackenzie-like DDF [71]; see Figure 8b.

Then, the GB energy and mobility are defined as being disorientation dependent, using a Read–Shockley (RS) [55] and a Sigmoidal (S) function [56], respectively:(12)$$\begin{array}{c} \left\{\begin{array}{c} \gamma \left(\theta \right)=\gamma_{max}\frac{\theta }{\theta_{0}}\left(1-ln\left(\frac{\theta }{\theta_{0}}\right)\right),\;\theta <\theta_{0}\\ \gamma_{max},\;\theta \geq \theta_{0} \end{array}\right. \end{array}$$
and
(13)$$\begin{array}{c} \mu \left(\theta \right)=\mu_{max}\left(1-exp\left(-5{\left(\frac{\theta }{\theta_{0}}\right)}^{4}\right)\right), \end{array}$$
where θ is the disorientation, $\mu_{max}$ and $\gamma_{max}$ are the GB mobility and energy of general high-angle grain boundaries (HAGBs). $\theta_{0}=$ 15° is the disorientation defining the transition from a low-angle grain boundary (LAGB) to a HAGB. The maximal value of GB energy is set to $\gamma_{max}=6\times 10^{-7}$ J · mm${}^{-2},$ which is typical for stainless steel [72,80]. The value of general HAGB mobility was computed using the methodology presented in Section 3.2 and is fixed at $\mu_{max}=0.476$ mm ${}^{4}\cdot$ J${}^{-1}$· s${}^{-1}$ for both isotropic and anisotropic formulations.

The simulations carried out using the anisotropic formulation consider heterogeneous GB energy defined by Equation (Equation 12) and two descriptions of the mobility. If GB mobility is isotropic, the formulation is referred as “Aniso(μ:Iso)”, and in the cases where GB mobility is heterogeneous (i.e., defined by Equation (Equation 13), the formulation is referred to as “Aniso(μ:S)”. Figure 9 shows the evolution of average quantities: normalized total GB energy $E_{\Gamma }/E_{\Gamma }\left(t=0\right)$, normalized number of grains $N_{G}/N_{G}\left(t=0\right)$, arithmetic mean grain radius ${\bar{R}}_{Nb}$, and normalized average GB disorientation $\bar{\theta }/\bar{\theta }\left(t=0\right)$. One can see that the mean grain radius evolution agrees with the experimental data and that the evolution of the other mean values are close to each other when using the different formulations and present reasonable variations from the experimental data. As stated in [22], the effect of a heterogeneous GB mobility does not affect the evolution of the mean values and distributions when orientations are generated randomly, and the DDF is similar to a Mackenzie distribution. One can see that the only mean value affected by the GB mobility is the average GB disorientation $\bar{\theta }$, being the “Aniso(μ:S)” formulation the one that is closer to the experimental evolution.

Figure 10 and Figure 11 show a good match of GSD and DDF between simulation results and experimental data after one and two hours of annealing at 1050 °C. One can see that the three cases are alike. The initial Mackenzie-like DDF evolves slowly for cases with random orientations and low anisotropy (as in [22,35,81,82,83]). Finally, in Figure 12, one can see the similarity between the microstructures obtained in the different simulations with most of the grains being equiaxed and few LAGBs.

The results presented here show that the evolution of an untextured polycrystal with an initial Mackenzie-like DDF could be simulated using an isotropic formulation or an anisotropic formulation with heterogeneous GB energy or both heterogeneous GB mobility and energy. This methodology has been used in different contexts under different annealing processes [72,75,76,77] and with academic microstructures in [22]. In the next section, the same analysis is performed by considering the same domain but by introducing special grain boundaries through an update of the μ and γ fields.

### 4.2. Statistical Case with an Improved Description of the γ and μ Fields

The microstructure used in this section, described in Figure 13, was generated using the same domain with L=2.0 mm and the GSD shown in Figure 7b. The initial number of grains is $N_{G}=14,956$ and their orientation is also generated randomly from the EBSD orientations. The initial number of grains is more important comparatively to the previous test case as the GSD described in Figure 2c, (left side) where Σ3 TBs are taken into account is used to generate the Laguerre–Voronoï polycrystal.

The same mesh and time step as the previous simulations are used in order to be able to fairly compare the obtained results. In order to define the behavior of special GBs with properties close to Σ3 TBs, the μ and γ fields are updated as follows:(14)$$\begin{array}{c} \left\{\begin{array}{cc} \gamma \left(\theta \right)=\gamma_{max}\frac{\theta }{\theta_{0}}\left(1-ln\left(\frac{\theta }{\theta_{0}}\right)\right), & \theta <\theta_{0}\\ \gamma \left(\theta \right)=\gamma_{max}, & \theta_{0}\leq \theta <\theta_{\Sigma 3}\\ \gamma \left(\theta \right)=\gamma_{max}*0.1, & \theta \geq \theta_{\Sigma 3} \end{array}\right. \end{array}$$
(15)$$\begin{array}{c} \left\{\begin{array}{cc} \mu \left(\theta \right)=\mu_{max}\left(1-exp\left(-5{\left(\frac{\theta }{\theta_{0}}\right)}^{4}\right)\right), & \theta <\theta_{\Sigma 3}\\ \mu \left(\theta \right)=\mu_{max}*0.1, & \theta \geq \theta_{\Sigma 3} \end{array}\right. \end{array}$$
with $\theta_{0}=$ 15° and $\theta_{\Sigma 3}=$ 60° and a value of GB energy and mobility set to $\gamma_{max}*0.1$ and $\mu_{max}*0.1$ for GBs with $\theta \geq \theta_{\Sigma 3}$. The value of $\mu_{max}$ is estimated using the evolution of ${\bar{R}}_{Nb[\%]}$ considering all GBs (see Figure 6). The estimated GB mobility is $\mu_{max}=0.069$ mm${}^{4}$· J${}^{-1}$· s${}^{-1}$, and it is one order of magnitude lower from the GB mobility estimated without Σ3 TBs. The decrease of the GB mobility is proportional to the decrease of grain size (see Figure 6) due to the high number of TBs.

From the results shown in Figure 14, one can see that all formulations minimize the energy with the same efficiency (Figure 14a) and the microstructures evolve at the same rate, leading to a good fit of mean grain size and number of grain evolutions comparatively to the experimental data. On the other hand, the anisotropic formulation shows a better agreement in terms of mean disorientation evolution.

Figure 15 shows the evolution of the grain size distribution at t=1 h and 2 h. GSDs present a good match (Figure 15) with small differences between the Iso and anisotropic formulations. However the DDF predictions (Figure 16) are quite bad for all formulations even if the anisotropic calculations tend to be better. This result can easily be explained by the use of statistics (GSD and orientations) from EBSD data which are not sufficient to accurately describe the real microstructure. First, the initial topology is simplified, but above all, even if the orientation data come from the EBSD measurements, the resulting initial DDF is not accurate, as a Mackenzie-like distribution is obtained, as illustrated in Figure 13b. Hence, the effect of the heterogeneous GB energy and the mobility anisotropy of the real microstructure are underestimated. A way of improvement of the proposed statistical generation methodology will be to modify the algorithm dedicated to the assignment of the orientation of each virtual grain by imposing also a better respect of the experimental DDF [84].

The GB energy of the microstructure at t=2 h is shown in Figure 17, a higher number of blue GBs, which correspond to TBs (low value of γ), are obtained using the anisotropic formulations, and the effect of heterogeneous GB mobility seems negligible.

The next simulations are carried out by immersing the EBSD data in order to overcome the limits discussed above.

## 5. Immersion of EBSD Data

In this section, a digital twin microstructure obtained by immersion of the EBSD map acquired on the initial microstructure (Figure 2) is discussed. Figure 18 shows the band contrast map of the microstructure and its numerical twin. The dimensions of the domain are $L_{x}=0.856$ mm and $L_{y}=1.138$ mm, and it contains 3472 crystallites. The time step is fixed at Δt=10 s, and the domain is discretized here using an unstructured static triangular mesh with a mesh size of h=1μm. This microstructure is more appropriate to compare simulations and experimental data. The evolution of the numerical microstructure is compared to EBSD maps obtained at three different times: t=30 min, 1 h, and 2 h (see Figure 3).

The GB energy and mobility are defined using Equations (Equation 14) and (Equation 15), respectively. The maximal value of GB energy is set to $\gamma_{max}=6\times 10^{-7}$ J · mm${}^{-2},$ and the maximal value of GB mobility is set to fit the mean grain size evolution. The maximal value of GB mobility for the Aniso(μ:Iso) and Aniso(μ:S) formulations are $\mu_{max}=0.146$ mm${}^{4}\cdot$ J${}^{-1}\cdot$ s${}^{-1}$ and $\mu_{max}=0.272$ mm${}^{4}\cdot$ J${}^{-1}\cdot$ s${}^{-1},$ respectively. Regarding the isotropic formulation, the value of GB reduced mobility remains constant $\mu \gamma =0.414\times 10^{-7}$ mm${}^{2}\cdot$ s${}^{-1}$. The changes in $\mu_{max}$ are due to the more complex geometry and the higher quantity of special boundaries that produce additional gradients of GB energy and GB mobility (see Equation (Equation 9)). As stated before, the additional jump at $\theta_{\Sigma 3}$ is set to define special boundaries similar to Σ3 TBs. Figure 19 confirms the good match between the TBs colored in red in the EBSD band contrast map (left side) and TBs colored in blue corresponding to a low GB energy in the numerical twin microstructure (right side).

With the immersed data, one can obtain a close digital twin of the real microstructure with the initial GB distributions presented in Figure 20. The initial GB energy distribution (GBED) is shown in Figure 20b. With this particular distribution, several junctions will have a high anisotropy level, and as stated in [22,83], one can expect different behaviors using the different formulations.

As illustrated in Figure 21, the three simulations predict similar trends concerning the mean grain size and the grain number evolution. Comparatively to experimental EBSD data, these predictions are very good concerning the mean grain size prediction, but all of them tend to predict, at the beginning, a faster disappearance of the small grains. Concerning the total energy, mean GB disorientation, and mean GB energy evolutions, the anisotropic formulation is closer to the EBSD data. This means that the Aniso formulation is more physical and promotes GBs with low GB energy.

Figure 22 illustrates the topology of grains at t=2 h. One can notice the higher quantity of GBs with low GB energy using the anisotropic formulation and its similarity to the EBSD band contrast map even if one can notice from the EBSD data that the real microstructure contains more TBs that create small grains as reflected in the GSD in Figure 23. Another advantage of the anisotropic formulation is the better reproduction of the DDF compared to the Iso formulation, which contains a lower percentage of GBs with $\theta \approx \theta_{\Sigma 3}$ and tends to promote a Mackenzie-like DDF (see Figure 24). One can also see in Figure 22, Figure 23 and Figure 24 that the heterogeneous GB mobility improves the morphology of grain, the GSD, and the DDF.

In this section, it has been shown that the immersed data give a better insight of the real microstructure evolution. In terms of mean values, there are small differences between the results from the Laguerre–Voronoï tesselation and the immersed microstructure. However, the GSD and DDF distributions are better reproduced for the immersed case when the anisotropic formalism is adopted. The heterogeneous GB mobility affects the grain topology, the GSD, and the DDF due to its additional retarding effect. Regarding the GSD, the Aniso(μ:S) formulation can reproduce the peak at low values of grain size and the peak around $\theta_{\Sigma 3}$ for the DDF, which are due to the TBs. Nevertheless, the behavior of TBs is still not perfectly reproduced by the proposed simulations, being the anisotropic formulation the one that seems more physical. In the next section, this issue is addressed using the GB5DOF code, which allows to define the GB energy in terms of misorientation and GB inclination (in 2D) in order to better characterize the evolution of TBs.

## 6. Using Anisotropic GB Energy and Heterogeneous GB Mobility

### 6.1. Simulation Results

In this section, the immersed polycrystalline microstructure and the FE mesh presented in Section 5 are used. Anisotropic GB energy values are defined using the GB5DOF code [6], and the heterogeneous GB mobility is described using Equation (Equation 15). It means that the GB energy can vary with the GB misorientation and inclination even if the torque terms are neglected. Note that the GB inclination is measured in 2D and not 3D; in other words, the GB is supposed to be perpendicular to the EBSD map. The GB energy of the microstructure and its GBED are shown in Figure 25. The initial microstructure is shown in Figure 25a, and one can see that the maximum value of GB energy is set around $\gamma_{max}\approx 7\times 10^{-7}$ J·mm${}^{-2}$. As discussed in [6], incoherent Σ3 TBs have a GB energy defined as $\gamma_{\Sigma 3}\approx 0.6*\gamma_{max}$, meaning that the modified Read–Shockley model described by Equation (Equation 14) seems exaggerated. In Figure 25b, the GBED is concentrated within the values $4\times 10^{-7}$ J·mm${}^{-2}\leq \gamma \leq 7\times 10^{-7}$ J·mm${}^{-2}$, which means that the level of heterogeneity is low and the different formulations are expected to promote similar trends, as stated in [22]. The GB mobility was set to fit the evolution of the mean grain size, the maximal GB mobility for the Aniso(μ:Iso) and Aniso(μ:S) formulations are, respectively, set to $\mu_{max}=0.0767$ mm${}^{4}\cdot$J${}^{-1}\cdot$ s${}^{-1}$ and $\mu_{max}=0.1423$ mm${}^{4}\cdot$ J${}^{-1}\cdot$ s${}^{-1}$. The difference between the $\mu_{max}$ values is generated by the higher values of GB energy produced by the GB5DOF code; note that with the RS model, all the TBs are defined as coherent, while the GB5DOF code can distinguish between coherent twin boundaries and incoherent twin boundaries, as pointed out in [22,38].

First, the mean grain size and grain size distribution evolutions are well reproduced by the different simulations (see Figure 26 and Figure 27). The mean GB disorientation is not well represented by any of the formulations, the EBSD data show a stable value around 50° while all numerical results exhibit a decreasing trend (see Figure 26b). This effect is due to the TBs and illustrates the inability for the numerical formulations to preserve or generate them. Additionally, the DDF from both formulations are similar and do not correspond to the experimental DDF (see Figure 28). The similarity between the isotropic and anisotropic simulations is due to the low anisotropy level, which may be produced by the lack of information of the GB inclination (see Figure 29). As stated in [22], when the GBED is concentrated around a specific value, both formulations can present a similar trend. This is confirmed with Figure 30, where a zoom on the GB network is shown at four different times, and one cannot see any obvious difference among the obtained microstructures with the three different simulations. The main difference of these results lies in the ability of the anisotropic formulation to keep more Σ3 TBs when the sigmoid description of μ is used.

### 6.2. Current State of the Modeling of 3D Anisotropic Grain Growth

A final question regarding the anisotropy of GB properties is still open: do the 3D descriptions of GB properties affect the microstructure evolution? Until now, most of the studies of GG in 3D have presented simulations of polycrystalline microstructures using different textures and a mathematical description of GB properties [33,50,51,52] or using databases of GB energy [53,54]. The following conclusions are pointed out:The effect of the heterogeneity is stronger when the material is textured or the disorientation transition between LAGBs and HAGBs, $\theta_{0}$, is high [51,52,54];The individual effect of GB energy and mobility is small on the GG [52].

Note that similar conclusions were presented in the first part of this work [22]. In [33,50,51,52], GB properties are defined as heterogeneous and not as anisotropic. The inclination dependency can have an important impact; hence, a complete description of the GB properties is necessary; i.e., $\mu (M\left(\theta ,\vec{a}\right),\vec{n})$ and $\gamma (M\left(\theta ,\vec{a}\right),\vec{n})$, as well as 3D non-destructive in situ characterization [45,47,48] in order to obtain more realistic values of GB mobility, must be considered as a key perspectives concerning the full-field modeling of GG.

In the simulations presented in this section, the GB inclination is simplified as it is projected in the observation plane. In other words, the description of the GB properties is simplified, and only a slice of the GB energy is considered. For instance, Figure 31 shows the GB energy and mobility of a Σ3 TB. One can see that the 3D surface of the TB properties has a complex geometry. On the other hand, the anisotropy of the GB mobility is simplified to a sigmoidal model with a cusp at $\theta_{\Sigma 3}$. Unluckily, the GB mobility data are not available for the complete GB space and at different temperatures.

## 7. Summary and Conclusions

Different FE-LS formulations to study the GG of 316L stainless steel were compared. The isotropic formulation is able to reproduce, for statistically generated or immersed polycrystals, the average grain size and grain size distribution for a wide range of anisotropy levels.

The results obtained using representative Laguerre–Voronoï polycrystals show that the heterogeneous GB mobility values do not affect the response of the different formulations and that the anisotropic formulation is more physical being the only formulation that enables promoting GBs with low energy. However, the anisotropy level was largely underestimated because of the initial Mackenzie-like DDF, which could not be controlled during the polycrystal generation.

Two additional cases were presented with a twin numerical microstructure obtained directly from EBSD data. The main advantage is that the initial DDF and topology are accurately defined. First, GB energy and mobility were defined using the modified Read–Shockley and sigmoidal models already tested for the virtual polycrystals. Then, the model was coupled with an anisotropic model of GB energy that takes into account the GB misorientation and inclination [6]. However, it is highlighted again that the GB inclination is not well defined, as the GB is supposed to be perpendicular to the observation plane. The proposed RS model seemed to exaggerate the anisotropy level comparatively to the GB5DOF code. However, the predictions are clearly better with the proposed RS and sigmoid models associated to the anisotropic formulation while not allowing to be predictive concerning the DDF whatever the method chosen.

These results illustrate that the prediction of grain growth at the polycrystal scale can be ambiguous depending the aimed attributes and the available data. First of all, 3D simulations should be considered. Of course, this aspect is, firstly, essential to improve the representation of the considered polycrystals but is also essential to describe correctly the γ dependence on the inclination. Indeed, the proposed 2D model/data context limits the actual use of the inclination, as this parameter is described here with one degree of freedom and not in a 3D framework with 3D experimental data. This aspect can explain the low anisotropy level obtained using the GB5DOF code. Finally, this objective must also be correlated to the fact to integrate the torque effects and thus the GB stiffness tensor in the simulations and analysis. It should be highlighted that this conclusion is common to all existing works of the state of the art involving anisotropic 2D GG simulations and 3D simulations where the inclination dependence, or torque terms, or both are not taken into account. Finally, it must also be emphasized that the influence of impurities segregation and/or oxides can be important for the considered alloy but was not discussed in the numerical framework; therefore, this constitutes another perspective of this work.

## Figures and Tables

**Figure 1 materials-15-02434-f001:**
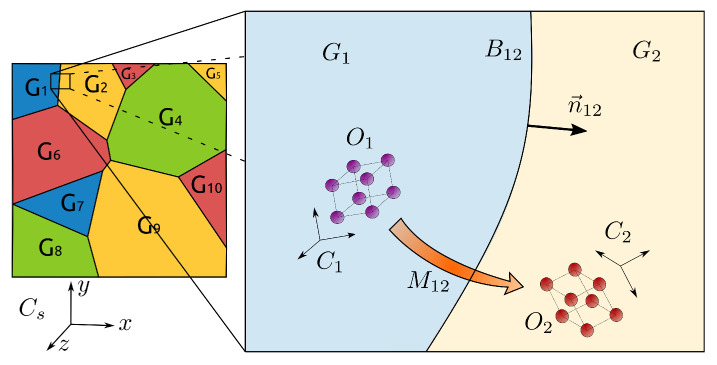
Scheme depicting one GB and its parameters. Image available online at Flickr (https://flic.kr/p/2m5JQkz, accessed on 15 June 2021) licensed under CC BY 2.0 (https://creativecommons.org/licenses/by/2.0/, accessed on 15 June 2021). Title: 10GGBParam. Author: Brayan Murgas.

**Figure 2 materials-15-02434-f002:**
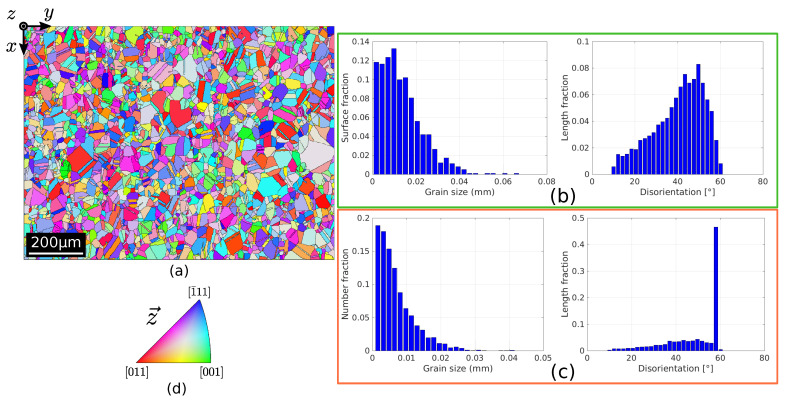
Initial microstructure properties determined by EBSD measurements. (**a**) IPF-z map. (**b**) Grain size distribution and DDF of grain boundaries excluding Σ3 TBs. (**c**) Grain size distribution as measured in 2D sections and DDF of grain boundaries including Σ3 TBs; the sharp peak on the DDF at 60° corresponds to Σ3 TBs. (**d**) Standard triangle used to color the orientation maps IPF-Z (indicating which crystallographic direction is lying parallel to the direction perpendicular to the scanned section).

**Figure 3 materials-15-02434-f003:**
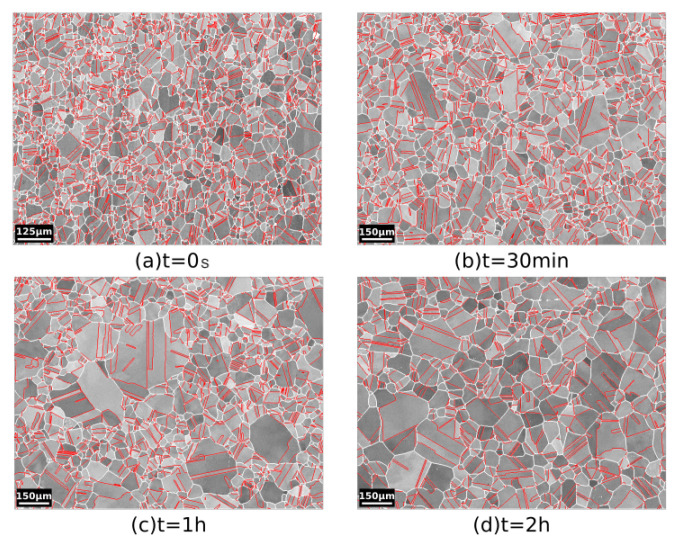
Annealing at 1050 °C: band contrast map of the microstructure of 316L steel at (**a**) t = 0 s, (**b**) t = 30 min, (**c**) t = 1 h, and (**d**) t = 2 h. Grain boundaries are depicted in white and Σ3 TBs are predicted in red.

**Figure 4 materials-15-02434-f004:**
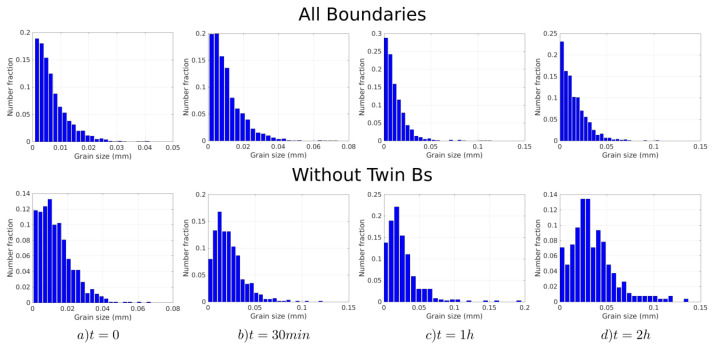
From left to right: evolution of the grain size histograms (in number) at t = 0 s, t = 30 min, t = 1 h, and t = 2 h. (**Top**): All boundaries are considered. (**Bottom**): Σ3 twin boundaries are excluded.

**Figure 5 materials-15-02434-f005:**
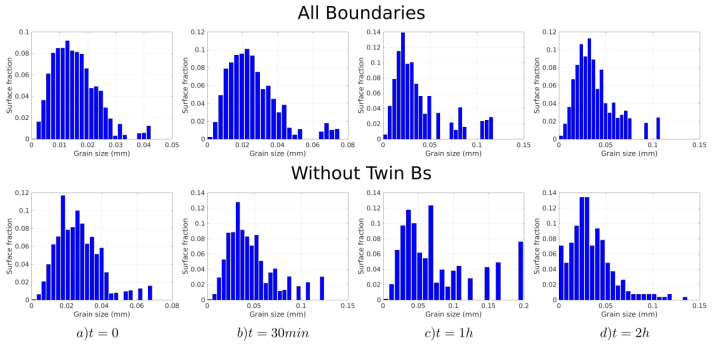
From left to right: evolution of the grain size histograms (in surface) at t = 0 s, t = 30 min, t = 1 h, and t = 2 h. (**Top**): All boundaries are considered. (**Bottom**): Σ3 twin boundaries are excluded.

**Figure 6 materials-15-02434-f006:**
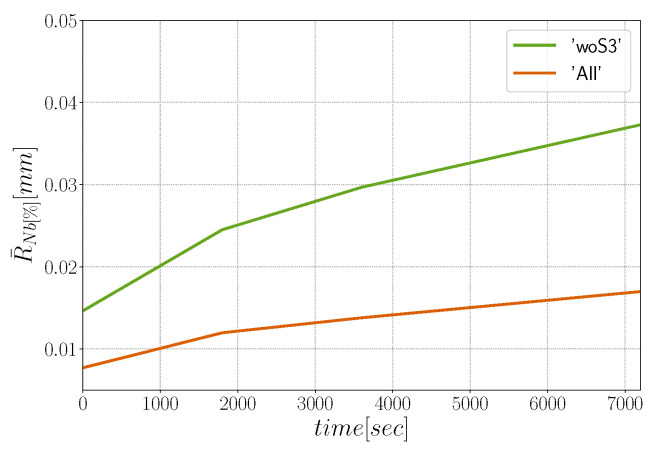
Mean grain radius evolution at 1050 °C from experimental data measured in 2D sections by taking into account all boundaries (in orange) and without TBs (in green). The outer grains that share a boundary with the image borders are not taken into account in the analysis.

**Figure 7 materials-15-02434-f007:**
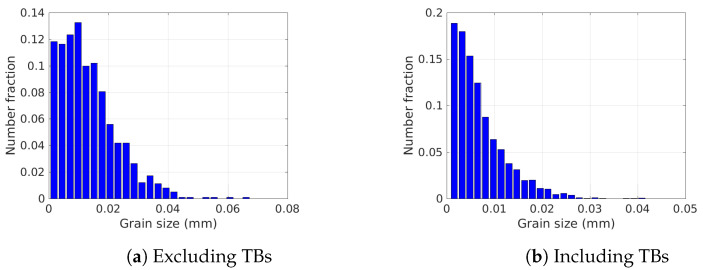
Initial grain size distributions (**a**) excluding TBs and (**b**) all grain boundaries obtained from the initial EBSD map shown in Figure 2.

**Figure 8 materials-15-02434-f008:**
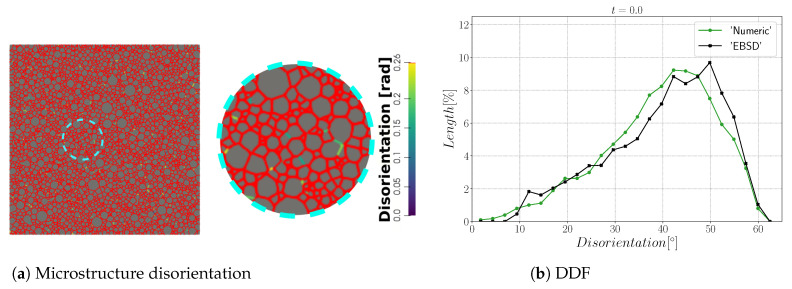
Initial (**a**) microstructure disorientation with a cyan circle that represents the zone shown on the right and (**b**) the disorientation distribution.

**Figure 9 materials-15-02434-f009:**
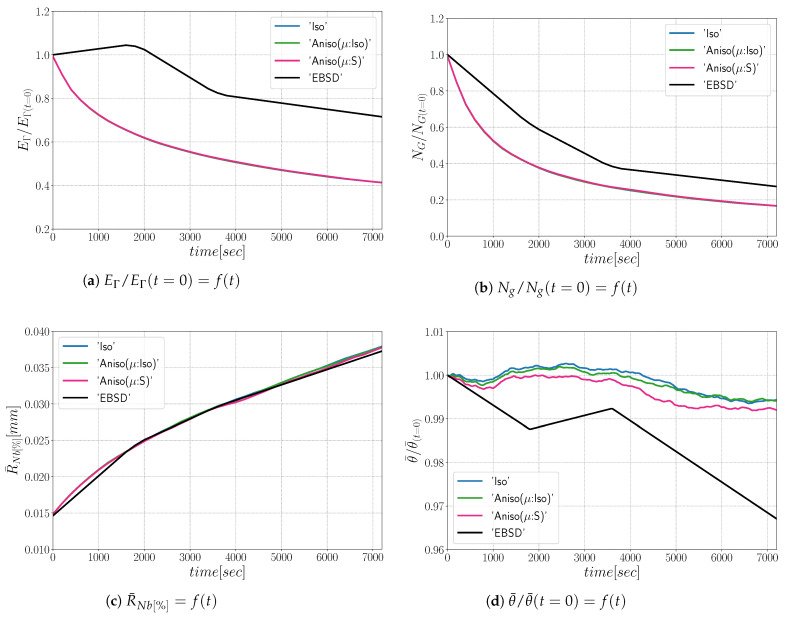
Mean values time evolution for the isotropic (Iso) formulation, anisotropic formulations with isotropic GB mobility (Aniso(μ:Iso)) and heterogeneous GB mobility (Aniso(μ:S)) and the experimental data (EBSD). Numerical results obtained from the initial microstructure shown in Figure 8a.

**Figure 10 materials-15-02434-f010:**
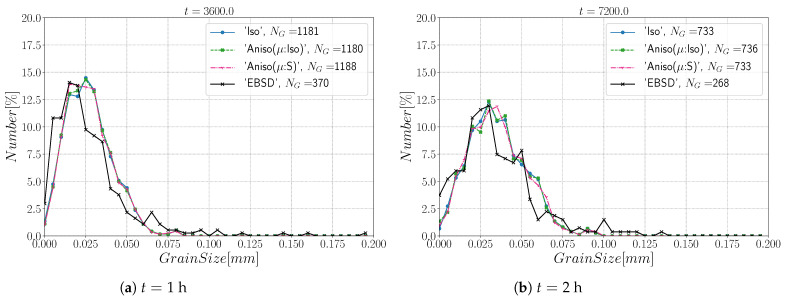
Grain size distributions obtained excluding TBs at (**a**) t = 1 h and (**b**) t = 2 h for the isotropic (Iso) formulation, anisotropic formulations with isotropic GB mobility (Aniso(μ:Iso)) and heterogeneous GB mobility (Aniso(μ:S)) and the experimental data (EBSD). $N_{G}$ refers to the number.

**Figure 11 materials-15-02434-f011:**
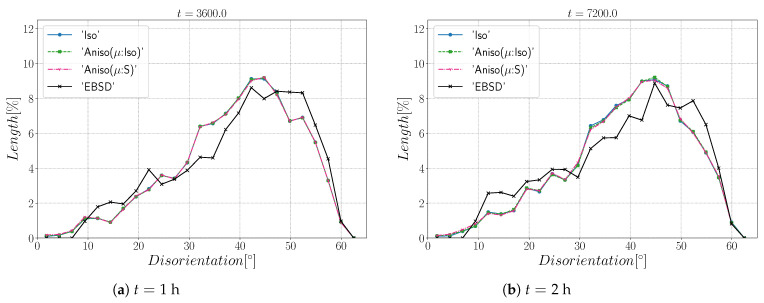
Disorientation distribution obtained excluding TBs at (**a**) t = 1 h and (**b**) t = 2 h for the isotropic (Iso) formulation, anisotropic formulations with isotropic GB mobility (Aniso(μ:Iso)), and heterogeneous GB mobility (Aniso(μ:S)) and the experimental data (EBSD). The y-axis represents the GB length percentage.

**Figure 12 materials-15-02434-f012:**
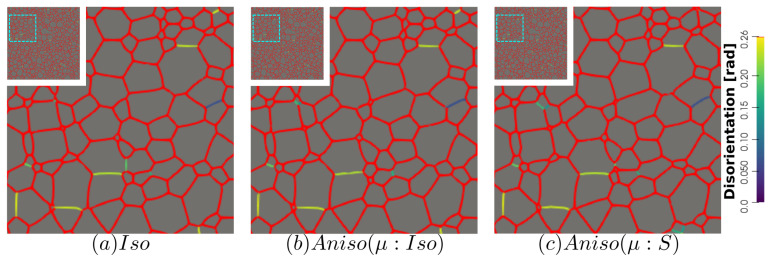
Detail of the GB disorientation at t=2 h in radians, GBs with a disorientation higher than 0.26 radians (15°) are colored in red: (**a**) isotropic framework, (**b**) anisotropic framework with γ function of θ (Equation (Equation 12)) and μ constant, and (**c**) anisotropic framework with γ and μ functions of θ through Equations (Equation 12) and (Equation 13), respectively. Due to the few GBs with θ<0.26, just a square section at the top-left of the hole microstructure is shown.

**Figure 13 materials-15-02434-f013:**
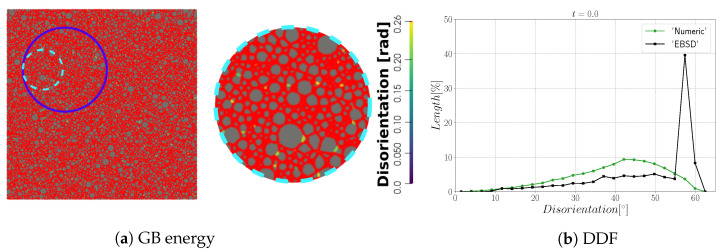
Initial (**a**) microstructure disorientation with a cyan circle that represents the zone shown on the right and (**b**) the disorientation distribution.

**Figure 14 materials-15-02434-f014:**
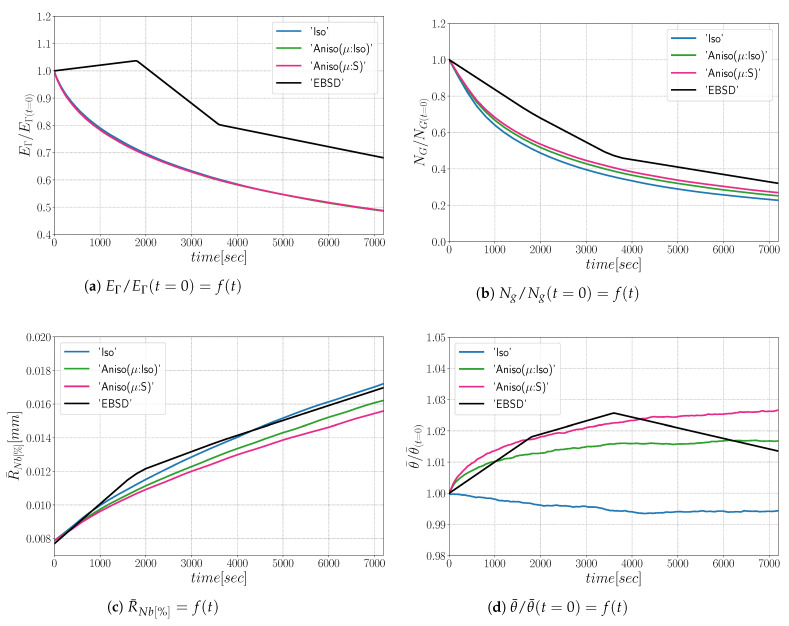
Mean values time evolution for the isotropic (Iso) formulation, anisotropic formulations with isotropic GB mobility (Aniso(μ:Iso)) and heterogeneous GB mobility (Aniso(μ:S)), and the experimental data (EBSD). Numerical results obtained from the initial microstructure shown in Figure 13a.

**Figure 15 materials-15-02434-f015:**
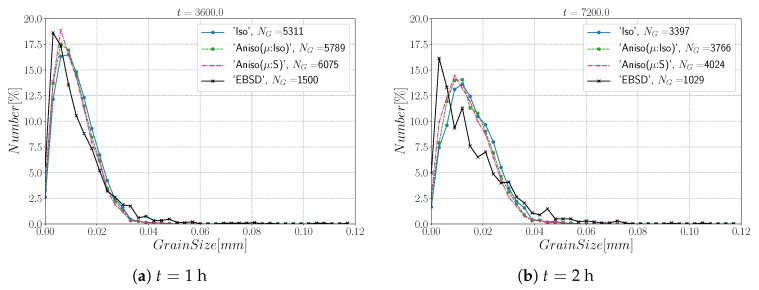
Grain size distributions obtained including TBs at (**a**) t = 1 h and (**b**) t = 2 h for the isotropic (Iso) formulation, anisotropic formulations with isotropic GB mobility (Aniso(μ:Iso)) and heterogeneous GB mobility (Aniso(μ:S)), and the experimental data (EBSD). $N_{G}$ refers to the number.

**Figure 16 materials-15-02434-f016:**
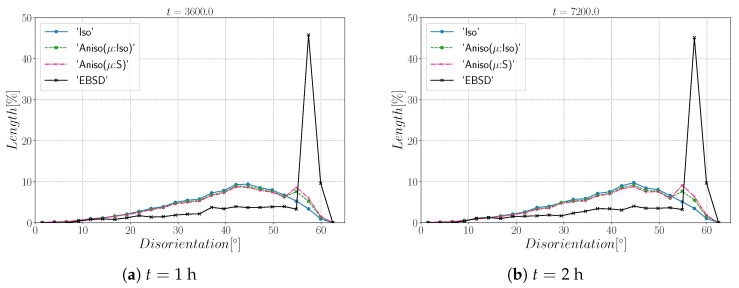
Disorientation distribution obtained including TBs at (**a**) t = 1 h and (**b**) t = 2 h for the isotropic (Iso) formulation, anisotropic formulations with isotropic GB mobility (Aniso(μ:Iso)) and heterogeneous GB mobility (Aniso(μ:S)), and the experimental data (EBSD). The y-axis represents the GB length percentage.

**Figure 17 materials-15-02434-f017:**
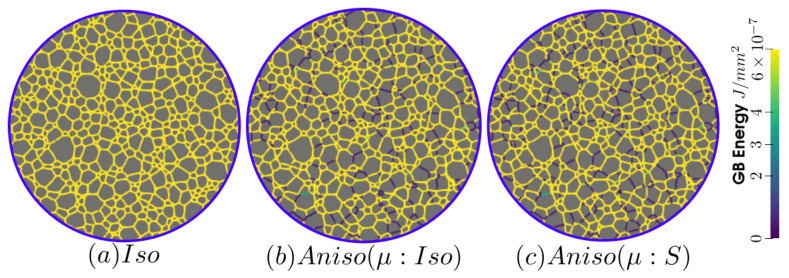
GB energy of the microstructure obtained with the (**a**) isotropic and anisotropic formulations using (**b**) isotropic GB mobility and (**c**) heterogeneous GB mobility at t=2 h in the same zone shown in Figure 13a.

**Figure 18 materials-15-02434-f018:**
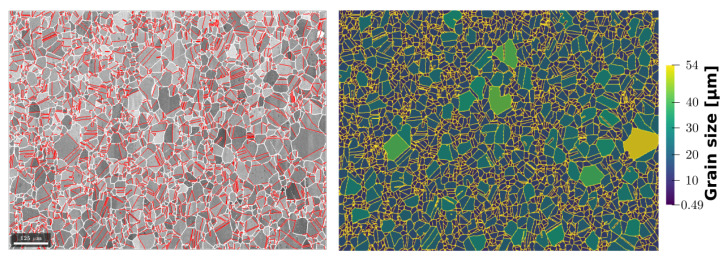
(**left**) EBSD band contrast map with GBs depicted in white and Σ3 TBs colored in red, and (**right**) its numerical microstructure displayed with a color code related to the grain size and GBs are colored in yellow. Here, TBs are considered to calculate grain size, i.e., crystallite size.

**Figure 19 materials-15-02434-f019:**
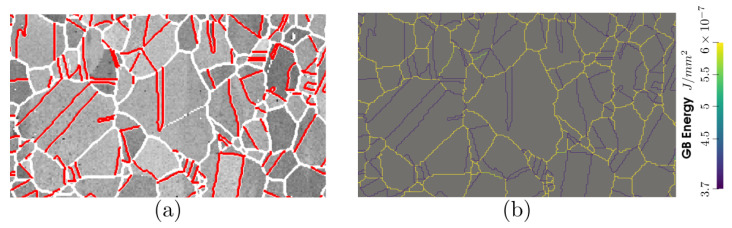
Detail of the (**a**) EBSD band contrast map and (**b**) its numerical twin showing the GB energy field. Twin boundaries depicted in red on the left image have low energy on the right image.

**Figure 20 materials-15-02434-f020:**
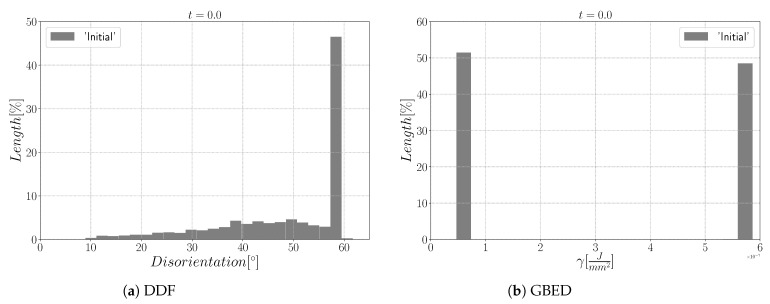
Initial DDF and GBED of the initial immersed microstructure produced by the modified Read–Shockley equation.

**Figure 21 materials-15-02434-f021:**
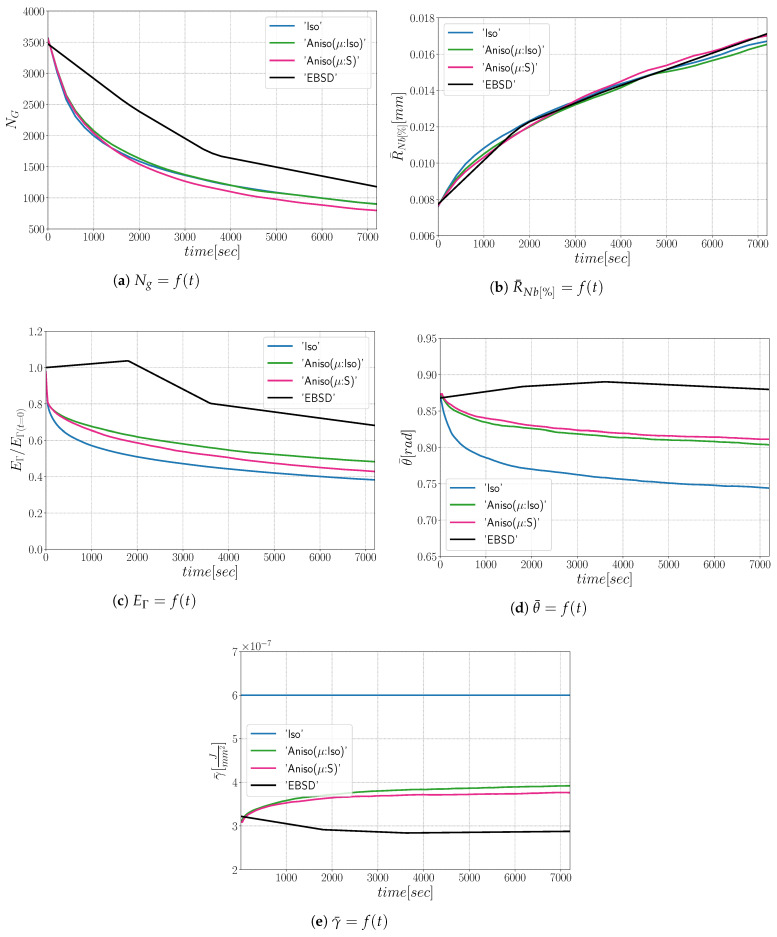
Mean values time evolution for the isotropic (Iso) formulation, anisotropic formulations with isotropic GB mobility (Aniso(μ:Iso)) and heterogeneous GB mobility (Aniso(μ:S)), and the experimental data (EBSD). Numerical results obtained from the initial microstructure shown in Figure 18.

**Figure 22 materials-15-02434-f022:**
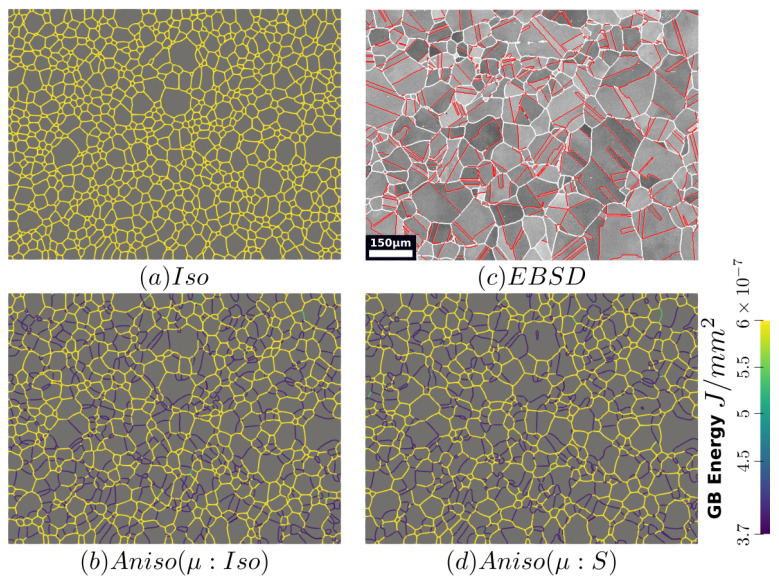
GB energy of the microstructures obtained numerically using the (**a**) isotropic formulation and the anisotropic formulations with (**b**) isotropic and (**d**) heterogeneous GB mobility. (**c**) illustrates the corresponding experimental band contrast map at t = 2 h.

**Figure 23 materials-15-02434-f023:**
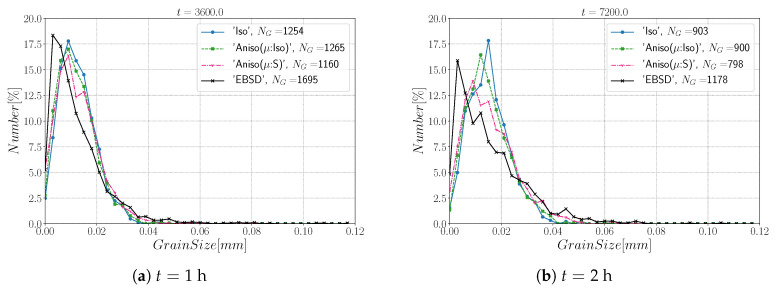
Grain size distributions obtained at (**a**) t = 1 h and (**b**) t = 2 h for the isotropic (Iso) formulation, anisotropic formulations with isotropic GB mobility (Aniso(μ:Iso)) and heterogeneous GB mobility (Aniso(μ:S)), and the experimental data (EBSD); $N_{G}$ refers to the number. Numerical results obtained from the initial immersed microstructure shown in Figure 18 and the RS and Sigmoidal model to define GB energy and mobility.

**Figure 24 materials-15-02434-f024:**
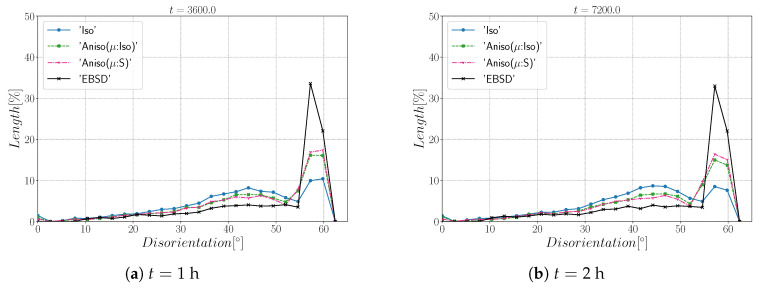
Disorientation distribution obtained at (**a**) t = 1 h and (**b**) t = 2 h for the isotropic (Iso) formulation, anisotropic formulations with isotropic GB mobility (Aniso(μ:Iso)) and heterogeneous GB mobility (Aniso(μ:S)), and the experimental data (EBSD). The y-axis represents the GB length percentage. Numerical results obtained from the initial immersed microstructure shown in Figure 18 and the RS and sigmoidal model to define GB energy and mobility.

**Figure 25 materials-15-02434-f025:**
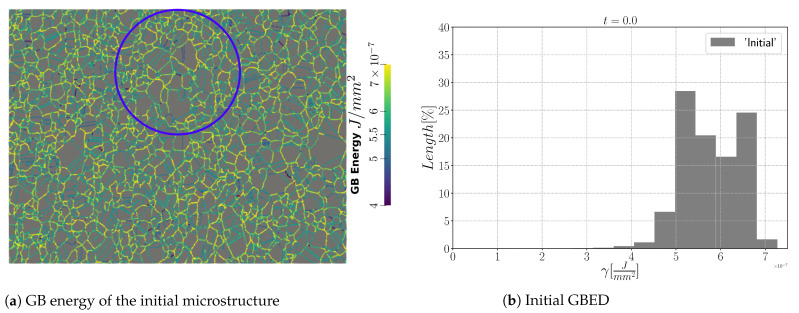
(**a**) GB energy field and (**b**) GBED of the initial immersed microstructure obtained using the GB5DOF code with the parameters $\epsilon_{RGB}=0.763$ J·mm${}^{-2}$ and AlCu-parameter = 0. In (**a**), the blue circle shows a zone of interest with a twin boundary composed of a coherent and incoherent part, similar to the one shown in [22].

**Figure 26 materials-15-02434-f026:**
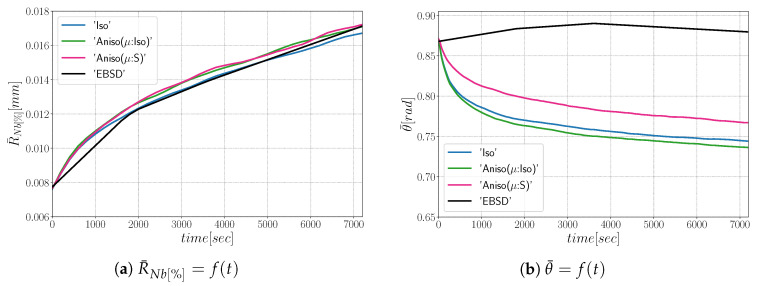
Mean values time evolution for the isotropic (Iso) formulation, anisotropic formulations with isotropic GB mobility (Aniso(μ:Iso)) and heterogeneous GB mobility (Aniso(μ:S)), and the experimental data (EBSD): (**a**) average grain radius, (**b**) average disorientation angle. Numerical results obtained from the initial immersed microstructure shown in Figure 25a and the GB5DOF code are used to define the GB energy.

**Figure 27 materials-15-02434-f027:**
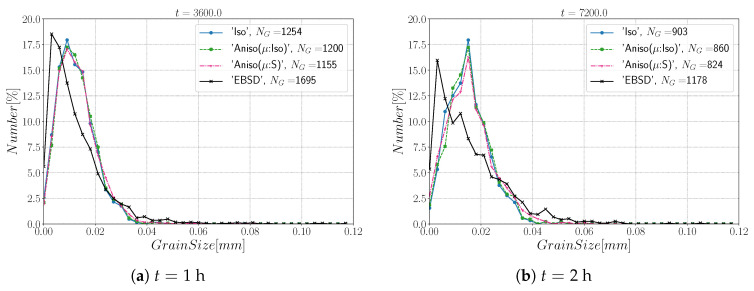
Grain size distributions obtained at (**a**) t = 1 h and (**b**) t = 2 h for the isotropic (Iso) formulation, anisotropic formulations with isotropic GB mobility (Aniso(μ:Iso)) and heterogeneous GB mobility (Aniso(μ:S)), and the experimental data (EBSD); $N_{G}$ refers to the number. Numerical results obtained from the initial immersed microstructure shown in Figure 25a and the GB5DOF code to define the GB energy.

**Figure 28 materials-15-02434-f028:**
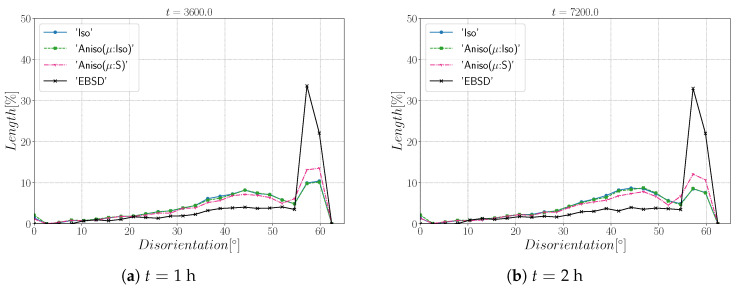
Disorientation distribution obtained at (**a**) t = 1 h and (**b**) t = 2 h for the isotropic (Iso) formulation, anisotropic formulations with isotropic GB mobility (Aniso(μ:Iso)) and heterogeneous GB mobility (Aniso(μ:S)), and the experimental data (EBSD). The y-axis represents the GB length percentage. Numerical results obtained from the initial immersed microstructure shown in Figure 25a and the GB5DOF code to define the GB energy.

**Figure 29 materials-15-02434-f029:**
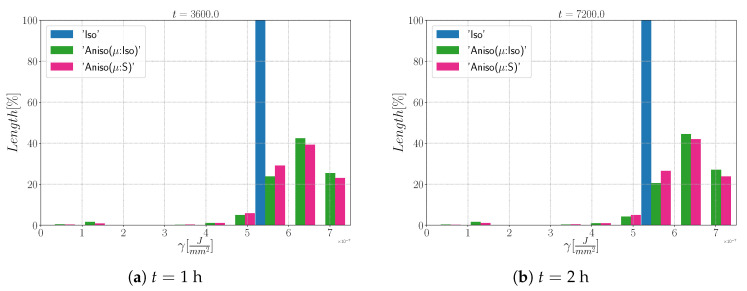
(**a**) t = 1 h and (**b**) t = 2 h for the isotropic (Iso) formulation and anisotropic formulations with isotropic GB mobility (Aniso(μ:Iso)) and heterogeneous GB mobility (Aniso(μ:S)). The y-axis represents the GB length percentage. Numerical results are obtained from the initial immersed microstructure shown in Figure 25a and the GB5DOF code to define the GB energy.

**Figure 30 materials-15-02434-f030:**
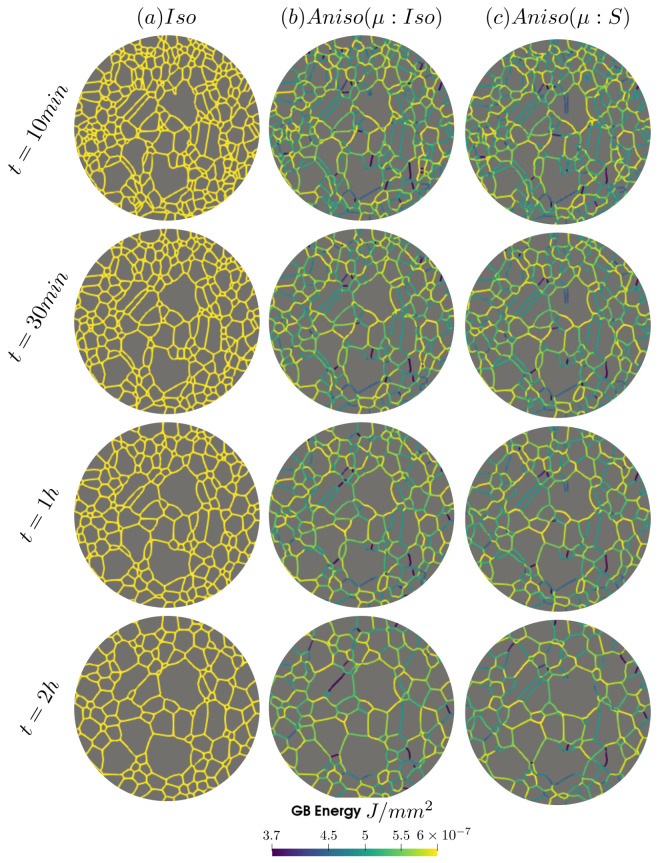
Microstructure evolution using the isotropic formulation and anisotropic formulation with isotropic and heterogeneous GB mobility at t=30 min, 1 h, and 2 h. The zone shown here is encircled in blue in Figure 25a.

**Figure 31 materials-15-02434-f031:**
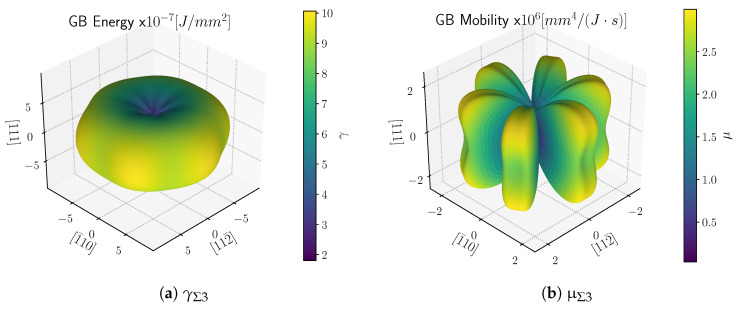
(**a**) GB Energy and (**b**) mobility of a Σ3 TB in Ni computed using the fits proposed in [67] of the atomistic simulation data in the study by Olmsted et al. [4,5]. The minimum, maximum, and average values of γ and μ are {1.803,6.793,10.064}$\times 10^{-7}$ J·mm${}^{-2}$ and {0.032,1.518,2.995}$\times 10^{6}$ mm${}^{4}\cdot$ J${}^{-1}\cdot$ s${}^{-1}$.

**Table 1 materials-15-02434-t001:** Chemical composition of the 316L stainless steel (weight percent).

Elem. Wgt%	Fe	Si	P	S	Cr	Mn	Ni	Mo	N
Min	bal.	-	-	-	16.0	-	10.0	2.0	-
Real	65.85	0.65	0.01	0.14	18.02	1.13	11.65	2.55	
Max	bal.	0.75	0.045	0.03	18.0	2.0	14.0	3.0	0.1

**Table 2 materials-15-02434-t002:** Polishing procedure applied to the 316L stainless steel samples. Plate and tower rate are the parameter of the used automatic polisher.

Abrasive	Time [s]	Plate [rpm]	Tower [rpm]	Force [dN]
320 SiC paper	60	250	150	2.5
600 SiC paper	60	250	150	2.5
1200 SiC paper	60	250	150	2.5
2400 SiC paper	60	150	100	1
HSV-3 μm Diamond	120	150	100	2
solution 0.12 mL/8 s				
electrolytic polishing	30 s	30 V	Electrolyte A2 (Struers)	

## Data Availability

The raw data required to reproduce these findings cannot be shared at this time as the data also form part of an ongoing study. The processed data required to reproduce these findings cannot be shared at this time as the data also form part of an ongoing study.

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
