# Peer review of "Level-Set Modeling of Grain Growth in 316L Stainless Steel under Different Assumptions Regarding Grain Boundary Properties"

_materials, 2022, doi:10.3390/ma15072434_

Round 1
Reviewer 1 Report
- “Based on Figures 3 and 4, the evolution of the microstructure seems to mostly proceed by normal grain growth (NGG) but the surface grain size distribution shows that the microstrucuture has a bimodal population of grains (see Figure 5).” Please explain it clearly
- “Most of the interfaces have a disorientation higher than 15° due to the random generation of orientations that leads to a Mackenzie-like DDF, see Figure 8b.” Most of the interfaces have a disorientation higher than 15°. Explain it, please!
- “The similarity between the isotropic and anisotropic simulations is due to the low anisotropy level, that may be produced by the lack of information of the GB inclination (see Figure29).” Please find out evidence to support this conclusion.
- Ref. [46] and [88] are PhD theses, which are not suit to be references of research paper.
Author Response
Dear editor, dear referees,
First, we would like to thank you for the attention you paid to our paper.
Answers, interwoven with each comment, are given below, and a few modifications have been realized in red and blue on the revised version of the paper.
Sincerely Yours,
The authors
Reviewer #1:
- “Based on Figures 3 and 4, the evolution of the microstructure seems to mostly proceed by normal grain growth (NGG) but the surface grain size distribution shows that the microstrucuture has a bimodal population of grains (see Figure 5).” Please explain it clearly
Figures 3 and 4 illustrate that globally no abnormal grain growth occurs (as the grain size is quite homogeneous without the appearance of rare very large grains), its why we can speak here about a normal grain growth evolution. However, it can also be highlighted that in terms of grain size distribution, one can see appearing slowly a bimodal shape (two main modes in the distribution).
- “Most of the interfaces have a disorientation higher than 15° due to the random generation of orientations that leads to a Mackenzie-like DDF, see Figure 8b.” Most of the interfaces have a disorientation higher than 15°. Explain it, please!
This strategy is quite usual in the state of the art. In fact, a Mackenzie-like DDF is equivalent to a random generation of the Euler angles which ensures a large majority of high angle grain boundary (disorientation higher than 15°). A reference was added.
- “The similarity between the isotropic and anisotropic simulations is due to the low anisotropy level, that may be produced by the lack of information of the GB inclination (see Figure29).” Please find out evidence to support this conclusion.
This aspect was already exhibited with simulations where high anisotropy levels are considered with a high impact concerning the LS results. This aspect is justified on the article just after this sentence and a reference is given.
- [46] and [88] are PhD theses, which are not suit to be references of research paper.
These references were canceled.

Reviewer 2 Report
This work studies the capacity of the isotropic and anisotropic formulations to model GG in a real material, 316L stainless steel, in terms of mean grain size, grain size distributions and mean GB properties. Compares the effect of the initial microstructure using statistically representative Laguerre-Voronoï tessellation and digital twin microstructures from EBSD data. The effect of the GB energy definition is compared with two different frameworks: 1-parameter and 5-parameter formulations.
The results using the isotropic and anisotropic formulations are compared using statistically representative Laguerre-Voronoï tessellations, immersed microstructures with heterogeneous GB properties and immersed microstructures with anisotropic GB energy using the GB5DOF code are very complete and very illustrative.
I have no corrections for this job.
Author Response
Dear editor, dear referees,
First, we would like to thank you for the attention you paid to our paper.
Answers, interwoven with each comment, are given below, and a few modifications have been realized in red and blue on the revised version of the paper.
Sincerely Yours,
The authors
Reviewer #2:
This work studies the capacity of the isotropic and anisotropic formulations to model GG in a real material, 316L stainless steel, in terms of mean grain size, grain size distributions and mean GB properties. Compares the effect of the initial microstructure using statistically representative Laguerre-Voronoï tessellation and digital twin microstructures from EBSD data. The effect of the GB energy definition is compared with two different frameworks: 1-parameter and 5-parameter formulations.
The results using the isotropic and anisotropic formulations are compared using statistically representative Laguerre-Voronoï tessellations, immersed microstructures with heterogeneous GB properties and immersed microstructures with anisotropic GB energy using the GB5DOF code are very complete and very illustrative.
I have no corrections for this job.
Thanks for your comments.

Reviewer 3 Report
In the presented work “Level-Set modeling of grain growth in 316L stainless steel under different assumptions regarding grain boundary properties”, the authors studied the models of growth and evolution of the grain structure with experimental comparison with the steel samples. The paper presents various approaches to predicting the evolution of grain boundaries with and without twin boundaries. The structure for modeling was based on grain sizes distribution and misorientations obtained by the EBSD method on a real sample. The grain structure evolution was controlled by comparing the parameters obtained by EBSD from samples annealed for different times. Despite the authors showed a significant effect of twin boundaries and used models with different anisotropy parameters, the experimental data only qualitatively correspond to the simulation. Perhaps the most developed models to date are not optimal. On the other hand, the cause could be the complex object considered in the experiment. Because it is obvious that the evolution of the grain structure is strongly influenced by impurity segregation and oxidation, which cannot be neglected in a real experiment with stainless steel.
Nevertheless, these remarks do not diminish the significance of the work, which has been excellently carried out. I recommend accepting the paper "Level-Set Modeling of grain growth in 316L stainless steel under different assumptions regarding grain boundary properties" for publication in Materials, after making minimal comments on the text.
1) In my opinion the paper looks rather difficult to understand. I would recommend the authors to put some information in the supplementary, it will improve the perception of the paper.
2) The description of experimental part of the paper could be improved. Describe the equipment and methodology of the experiment in more detail.
3) Check the text, section labels and graphs for compliance with the requirements of the journal.
Author Response
Dear editor, dear referees,
First, we would like to thank you for the attention you paid to our paper.
Answers, interwoven with each comment, are given below, and a few modifications have been realized in red and blue on the revised version of the paper.
Sincerely Yours,
The authors
Reviewer #3:
In the presented work “Level-Set modeling of grain growth in 316L stainless steel under different assumptions regarding grain boundary properties”, the authors studied the models of growth and evolution of the grain structure with experimental comparison with the steel samples. The paper presents various approaches to predicting the evolution of grain boundaries with and without twin boundaries. The structure for modeling was based on grain sizes distribution and misorientations obtained by the EBSD method on a real sample. The grain structure evolution was controlled by comparing the parameters obtained by EBSD from samples annealed for different times. Despite the authors showed a significant effect of twin boundaries and used models with different anisotropy parameters, the experimental data only qualitatively correspond to the simulation. Perhaps the most developed models to date are not optimal. On the other hand, the cause could be the complex object considered in the experiment. Because it is obvious that the evolution of the grain structure is strongly influenced by impurity segregation and oxidation, which cannot be neglected in a real experiment with stainless steel.
Thanks for this remark. This comment was added on the conclusion.
Nevertheless, these remarks do not diminish the significance of the work, which has been excellently carried out. I recommend accepting the paper "Level-Set Modeling of grain growth in 316L stainless steel under different assumptions regarding grain boundary properties" for publication in Materials, after making minimal comments on the text.
Thanks a lot for this positive comment.
1) In my opinion the paper looks rather difficult to understand. I would recommend the authors to put some information in the supplementary, it will improve the perception of the paper.
In the collective opinion of the authors, we found it difficult to define what we could add to the supplementary without making the article more cumbersome overall or less clear. We are of course open to suggestions.
2) The description of experimental part of the paper could be improved. Describe the equipment and methodology of the experiment in more detail.
The section 3 was slightly improved. In the author opinions, this section contains all the information needed (equipment and procedure) to understand/reproduce the experiments which are otherwise quite standard. Of course, we remain at the disposal of the reviewer for any information that he/she would consider missing in this part.
3) Check the text, section labels and graphs for compliance with the requirements of the journal.
Done.

Round 2
Reviewer 1 Report
Accept